# The Effect of Staging Intervals on Progression-Free Survival in Registration Studies of Oncologic Drugs: A Meta-Analysis

**DOI:** 10.3390/cancers17081359

**Published:** 2025-04-18

**Authors:** Jonas A. Zuellig, Roman Adam, Filomena Udry, Ariadna Tibau, Bostjan Šeruga, Alberto Ocaña, Eitan Amir, Arnoud J. Templeton

**Affiliations:** 1Faculty of Medicine, University of Basel, 4001 Basel, Switzerland; jonas.zuellig@stud.unibas.ch (J.A.Z.); roman.adam@unibas.ch (R.A.); filo.udry@stud.unibas.ch (F.U.); 2Oncology Department, Hospital de la Santa Creu i Sant Pau, Institut d’Investigació Biomèdica Sant Pau, 08041 Barcelona, Spain; atibau@santpau.cat; 3Departament de Medicina, Universitat Autònoma de Barcelona, 08041 Barcelona, Spain; 4Department of Medical Oncology, Institute of Oncology Ljubljana, 1000 Ljubljana, Slovenia; bseruga@onko-i.si; 5Faculty of Medicine, University of Ljubljana, 1000 Ljubljana, Slovenia; 6Experimental Therapeutics Unit, Medical Oncology Department, Hospital Clínico Universitario San Carlos IdISSC, 28040 Madrid, Spain; albertocana@yahoo.es; 7Instituto de Investiagción Sanitaria del Hospital Clínico San Carlos, 28040 Madrid, Spain; 8Division of Medical Oncology and Hematology, Department of Medicine, Princess Margaret Cancer Center, Toronto, ON M5S 1A1, Canada; eitan.amir@uhn.ca; 9Department of Medicine, University of Toronto, Toronto, ON M5S 1A1, Canada; 10Department of Medical Oncology, St. Claraspital, 4058 Basel, Switzerland; 11St. Clara Research, 4058 Basel, Switzerland

**Keywords:** meta-analysis, oncologic drug, registration trial, staging

## Abstract

This analysis investigated whether the frequency of radiographic assessments affects the perceived effectiveness of clinical trials for cancer drugs. Progression-free survival (PFS), a term used to define the time a patient lives without disease progression, was used for the analysis. This study analyzed pivotal studies supporting drug approvals in Switzerland from 2010 to 2022. Their findings showed that shorter intervals between scans (less than 8 weeks) were linked to higher hazard ratios (HRs), meaning a lower apparent treatment effect, while longer intervals (8 weeks or more) showed a stronger effect. This puts the concern that frequent scans might exaggerate drug benefits into a new perspective. However, results varied by cancer type, drug type, and also the primary outcome of the studies. This study had limitations, such as relying on published rather than individual patient data. Although these findings suggest that the timing of disease assessments may influence trial outcomes, definitive answers require more research to better understand the potential impact of staging intervals.

## 1. Introduction

In recent years, a substantial number of new drugs for the treatment of metastatic cancer have been developed and approved based on surrogate endpoints [1]. Increasingly, progression-free survival (PFS) has been adopted as a primary outcome measure, which is usually assessed by regular radiographic imaging based on standardized criteria like RECIST [2,3,4,5]. In clinical studies PFS is usually defined as the time from randomization to the radiographic documentation of progression or death. The date of the radiologic evaluation at which progression is first evident is thereby used as a proxy for the true progression time, since the true time of progression typically lies between the two assessments [6]. This leads to an overestimation of the true PFS, and an apparently longer median PFS may just be a consequence of the length of the surveillance interval [7,8,9]. In 2021, Dabush and colleagues reported that in clinical studies of metastatic breast cancer shorter staging intervals (<9 weeks) are associated with lower hazard ratios (HRs) compared to studies applying longer intervals for restaging and thus suggesting a greater treatment effect in terms of PFS [10].

The aim of our work was to explore the potential impact of restaging intervals in studies leading to the registration of oncologic drugs in Switzerland to corroborate or to put into perspective the hypothesis that shorter staging intervals are associated with apparent higher magnitude of the PFS effect across various disease sites.

## 2. Materials and Methods

### 2.1. Data Sources and Searches

This analysis was conducted in line with the Preferred Reporting Items for Systematic Reviews and Meta-Analyses (PRISMA) guidelines [11]. New drugs and new indications for earlier approved drugs for market authorization in Switzerland between 2010 and 2022 were identified from the official Journal of Swissmedic, the national authorization and supervisory authority for drugs and medical products, and its website swissmedicinfo.ch [12]. Subsequently, studies supporting the authorization of the respective drug and indication were searched based on the information available with the official label [13].

### 2.2. Study Selection

The following selection criteria were used: (i) pivotal study supporting drug registration with PFS as primary or secondary endpoint, (ii) availability of HR for PFS with corresponding 95% confidence intervals (CIs) and/or *p*-value, and (iii) staging interval of radiographic assessment reported. Studies in the curative setting (i.e., neo-/adjuvant therapies) and the pediatric setting and of non-oncologic drugs (e.g., for supportive treatment) or hematologic indications (e.g., myeloma, lymphoma, and leukemia) were not included.

### 2.3. Data Extraction

Data were collected using predesigned abstraction forms. The following data were extracted from original publications: name of first author, year of publication, study phase and design, disease site, drug class, indication (e.g., line of therapy), staging interval, primary endpoint(s), HR for PFS, and associated 95% CI and/or *p*-value. HRs were preferably extracted from multivariable models where available. For studies with time-varying staging intervals (e.g., longer intervals with longer follow-up), the initial staging intervals were chosen. In the case of studies which included patients in various lines of treatment (e.g., first-line and second-line) these were classified as “other line”.

When the indication for authorization corresponded to a specific subgroup evaluated within a study, data for that subgroup were used if reported separately [14]. If the indication was based on multiple studies with differing selection criteria—such as mutation status, prior therapies, or treatment lines—all relevant studies were included in the evaluation. When the indication was broadened to include additional subgroups—such as a new mutation class, treatment line, or an extended authorization across different treatment stages—the corresponding study data supporting the expansion were reviewed. If multiple studies with identical selection criteria supported the authorization, the study with the largest sample size was chosen. In trials reporting multiple subgroups, the subgroup most closely aligned with the approved indication was selected.

### 2.4. Data Synthesis

HRs were pooled in a meta-analysis. The prespecified primary analysis was dichotomized according to the median staging interval (<median vs. ≥median). Subsequently, subgroup analyses according to drug classes and disease sites were carried out if a subgroup consisted of at least 5 studies. Exploratory subgroup analyses were performed according to the trial phase and according to the primary outcome in the respective studies. Subsequently, sensitivity analyses using staging interval cut offs other than the median and only including phase 3 studies, were performed. To further address heterogeneity and to test the robustness of the findings the analyses were repeated after the exclusion of outliers (i.e., studies with staging intervals < 6 weeks or >12 weeks) [15].

### 2.5. Statistical Analyses

Data were combined into a meta-analysis using RevMan 5.4 analysis software. [16] Estimates of HR were weighted and pooled using the generic inverse-variance and random-effect model [17]. Differences between the subgroups were assessed using methods described by Deeks et al. [18]. Heterogeneity was assessed using Cochran Q and I^2^ statistics [19]. To explore the potential impact of staging intervals as a continuous variable, a meta-regression was performed with the staging interval as the independent variable and the natural logarithm of the HR (Ln (HR)) as the dependent variable utilizing SPSS version 25.0 (IBM Corp. Armonk, NY, USA) [20]. Weighting was performed with the inverse of the variance of the HR. Publication bias was not formally assessed since all studies included in the analysis supported drug approvals, and thus sufficient quality was assumed. All statistical tests were two-sided, and statistical significance was defined as *p* < 0.05. No correction was made for multiple significance testing.

## 3. Results

### 3.1. Included Studies

Between 2010 and 2022 73 drugs for 167 indications received market authorization by Swissmedic (Figure 1).

In total, 112 studies met the selection criteria and were included in the analysis. The study characteristics are given in Table 1. Most studies were randomized phase 3 studies (93%); the most common disease sites were lung cancer, breast cancer, and gastrointestinal malignancies. Immunotherapies and targeted therapies (small molecules) were the largest groups of drug classes. PFS was the most commonly used primary or co-primary endpoint.

### 3.2. Staging Intervals

The median staging interval was 8 weeks (range 4–18). Fifty-one studies (46%) had staging intervals < 8 weeks and sixty-one (54%) had staging intervals ≥ 8 weeks (Table 1). The pooled HR for staging intervals less than the median was 0.58 (95% CI 0.53–0.64), while the pooled HR for staging intervals equal or longer than the median was 0.48 (95% CI 0.44–0.52) with a *p*-value of 0.005 for the subgroup difference. There was significant statistical heterogeneity (I^2^ = 90%, *p* < 0.001) which could not be explained by outlier studies (Figure 2).

### 3.3. Subgroup Analyses

The differences between the subgroups according to drug classes and disease sites are shown in Table 2. No significant difference between the pooled HRs for PFS among the different drug classes was observed for restaging intervals < 8 weeks compared to restaging intervals ≥ 8 weeks. In studies of melanoma, shorter staging intervals were associated with a lower pooled HR (the HR for staging intervals < 8 weeks was 0.46 vs. 0.63 for staging intervals ≥ 8 weeks, *p* = 0.02), whereas in studies of renal cell cancer the opposite was observed, i.e., longer staging intervals were associated with a lower pooled HR (the HR for staging intervals < 8 weeks was 0.67 vs. 0.44 for staging intervals ≥ 8 weeks, *p* = 0.01). In all other tested subgroups according to disease sites there was no significant difference. In the subgroups according to the trial phase, the overall finding remained unchanged in phase 3 studies and did not reach a statistical significance in the fairly small group of phase 2 studies. Interestingly, when grouping the studies according to their primary outcome, the finding of lower HRs with longer staging intervals was only observed in studies with OS as the primary outcome (0.72 vs. 0.58, *p* = 0.03).

### 3.4. Sensitivity Analyses

Sensitivity analyses were performed to control the cut-off effects of the staging intervals on the outcomes. With a cut-off of 9 weeks (i.e., <9 weeks vs. ≥9 weeks), as in the work by Dabush et al., we found a similar effect as with the median of 8 weeks, namely a numerically higher pooled HR for staging intervals < 9 weeks compared to staging intervals ≥ 9 weeks (HR 0.54 vs. 0.45, *p* for subgroup difference 0.06). Similar results were found for cut-offs at 6 and 12 weeks and the cut-off at 12 weeks (Table 3). When studies with outliers (i.e., staging interval < 6 weeks or >12 weeks) were excluded, the main result remained unchanged (HR for staging interval < 8 weeks vs. ≥8 weeks 0.60 vs. 0.49, *p* for subgroup difference = 0.002). Further sensitivity analyses after the exclusion of phase 2 studies yielded similar results (Appendix A). Yet, significant heterogeneity remained and could not be explained by the removal of single studies.

### 3.5. Meta-Regression

Evaluating the potential effect of the staging interval as a continuous variable on the HR for PFS, a meaningful correlation of lower HRs with longer staging intervals was observed based on the Burnand criteria [22] (beta −0.422; *p* < 0.001). In a sensitivity analysis including phase 3 trials only, similar results were found (beta −0.427, *p* < 0.001).

## 4. Discussion

Earlier reports suggest that, when assessing PFS in studies of metastatic breast cancer, shorter restaging intervals are associated with lower HRs and might thus bias the conclusion of the apparent benefit of experimental drugs. This prompted us to explore this issue in registration studies of oncologic drugs across various disease sites during a 13-year period. In this analysis, overall shorter staging intervals were associated with higher HRs (i.e., lower effect size). This finding is reassuring since it puts any claim into perspective that by selecting shorter staging intervals the PFS results of a clinical study might be biased in favor of the experimental arm. Yet, how might the different findings be explained? First, Dabush et al. pooled 98 studies in metastatic breast cancer, while in our study there was considerable clinical heterogeneity, explaining the overall high statistical heterogeneity [18]. Also, the tumor type and underlying heterogenous biology may explain the divergence of findings in the indication subgroup analyses. As tumors with a more aggressive biology exhibit a shorter time to progression [23], they may be assessed more frequently with shorter staging intervals [10,24,25,26]. Since changes in PFS during brief intervals could have the tendency to be more modest, it might influence the hazard ratios observed in studies using shorter staging intervals [10]. Second, the difference in the pooled HRs reported by Dabush et al. was quite small and likely not clinically meaningful (HR 0.79 vs. 0.86, *p* for subgroup difference 0.15); however, significant findings were observed in non-first-line trials, trials with drugs replacing standard treatments, and studies performed exclusively in human epidermal growth factor receptor 2 (HER2)-positive disease. Our study only included 18 studies supporting drug authorization in breast cancer, which did not allow us to comprehensively assess the respective subgroups with adequate power. Notably, in line with earlier findings, we found numerically lower HRs with shorter staging intervals (0.50 vs. 0.58, *p* for subgroup difference = 0.28) in breast cancer trials, but these differences are likely non-meaningful.

Our work has several limitations which should be considered, and open questions remain. First, the analysis was based on published HRs rather than the individual patient data needed for more in-depth analyses [27,28]. Second, we considered pivotal studies highlighted in the initial drug label with the authorization of a drug in a specific indication. Earlier and other studies also supporting drug authorization may thus have been missed, decreasing our ability and the power of our analysis to detect the potential effects of staging intervals on the magnitude of the PFS benefit—especially in smaller subgroups of drug classes and disease sites [29]. Third, there was considerable heterogeneity which could not be adequately explained despite multiple subgroup and sensitivity analyses. This raises the question of whether some findings might be due to chance, e.g., in the subgroup of studies for melanoma where shorter staging intervals were associated with apparent greater effects on PFS without an obvious biological explanation. Fourth, the main finding reported here appears to be driven by studies with OS as the primary outcome, where staging intervals only influence secondary outcomes. The reason for this remains unclear and should be further explored, although the practical relevance seems limited since OS is clearly the more important outcome for patients in comparison to the PFS measure, and the influence of staging intervals on OS appears unlikely. Fifth, we did not have the adequate power to perform multivariable meta-regression analyses to control for factors potentially influencing the observed correlation of longer staging intervals with lower HRs. Sixth, the basis of our analysis was drug authorizations in Switzerland, which have their own regulatory body. Thus, the results might be slightly different when considering authorizations by the United States’ Food and Drug Administration (FDA) or the European Medicines Agency (EMA). Although Swissmedic tends to grant the market access of drugs in the (neo-)adjuvant setting more restrictively than its counterparts, this is less so in the palliative setting assessed here [30,31]. Thus, the results provided here are likely generalizable to other jurisdictions [32]. Seventh, we did not include studies supporting approvals of drugs used in pediatric oncology and hematologic malignancies, leaving the potential impact of staging intervals in such studies unexplored. Last but not least, the analyses were all univariable which makes it difficult to know how many of the findings are independent [33,34].

## 5. Conclusions

In conclusion, in the studies leading to the authorization of oncologic drugs in the palliative setting, longer rather than shorter restaging intervals to measure PFS were associated with an apparent higher magnitude of effect. This observation was the opposite of what was anticipated. Thus, the potential impact of staging intervals on PFS outcomes in randomized studies, considering disease biology, warrants further research.

## Figures and Tables

**Figure 1 cancers-17-01359-f001:**
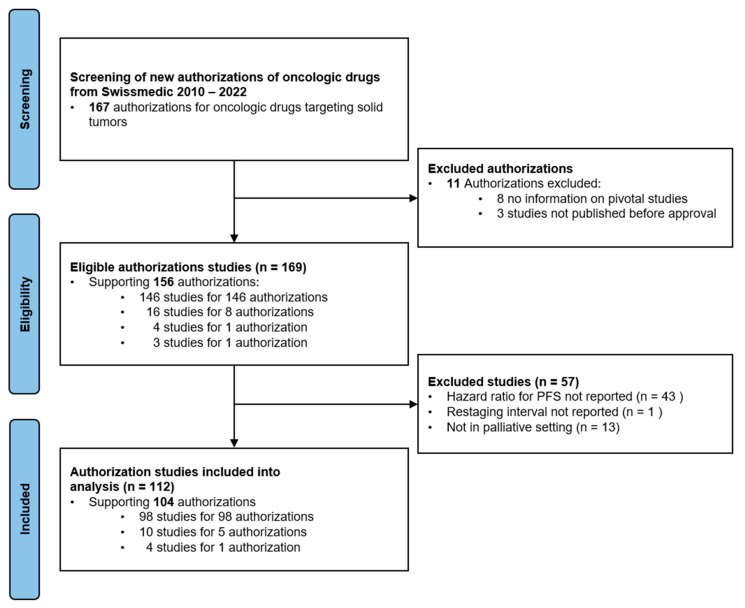
Studies included in analysis (PRISMA flow diagram [21]).

**Figure 2 cancers-17-01359-f002:**
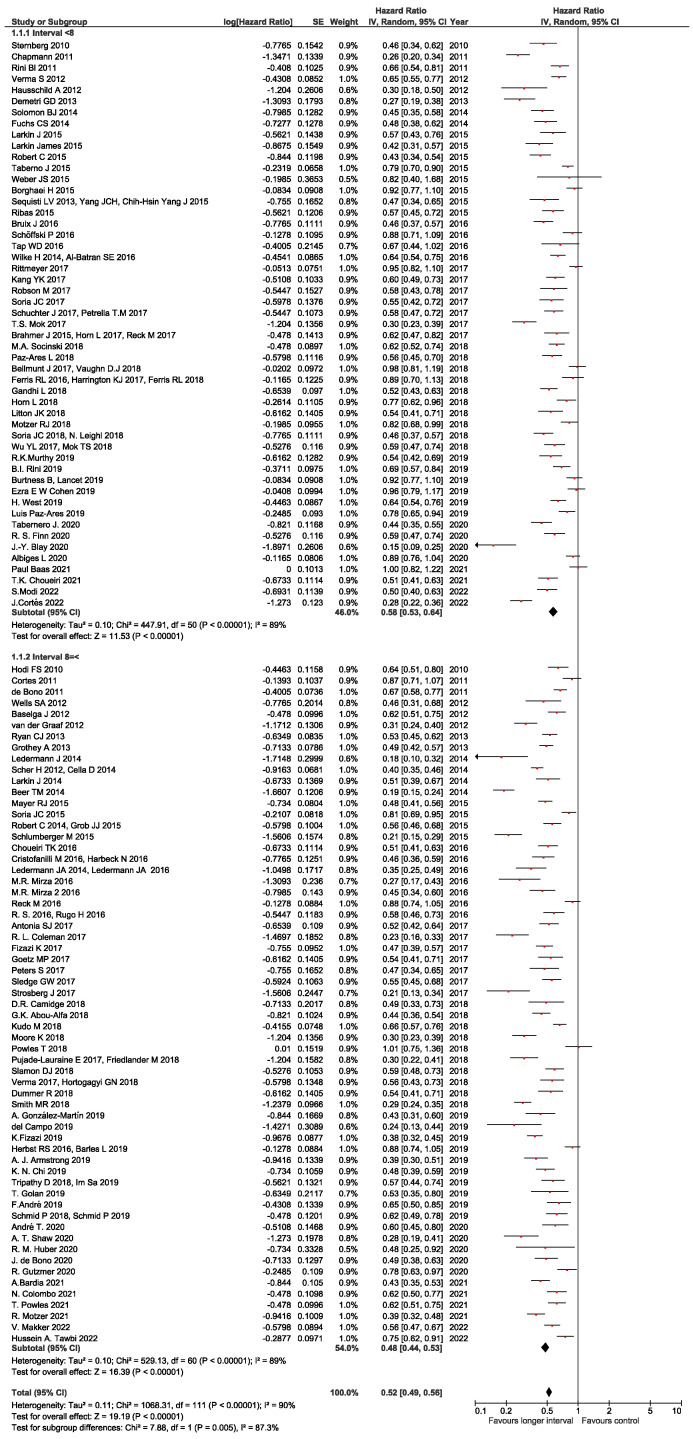
Forest plots showing hazard ratio for PFS less than versus greater or equal than median staging interval (8 weeks). Hazard ratios (HRs) for each study are represented by squares; size of square represents weight of study in meta-analysis, and horizontal line crossing square represents 95% confidence interval (CI). All statistical tests were two-sided.

**Table 1 cancers-17-01359-t001:** Characteristics of included studies.

	Total (n = 112)	Restaging Interval < 8 weeks (n = 51)	Restaging Interval ≥ 8 weeks (n = 61)
Sample Size (Median, IQR)	531 (342–713)	537 (351–658)	525 (341–762)
Primary Outcome			
PFS, n (%)	55 (49%)	18 (16%)	37 (3%)
OS, n (%)	25 (22%)	14 (12%)	11 (10%)
PFS and OS n (%)	25 (22%)	15 (13%)	10 (9%)
Others n (%)	7 (6%)	4 (3%)	3 (3%)
Trial Phase			
Phase 3 Trial, n (%)	105 (93%)	48 (43%)	57 (50%)
Phase 2, n (%)	6 (5%)	3 (3%)	3 (2%)
Phase 2 and 3 (%)	1 (1%)	-	1 (1%)
Randomized, n (%)	112 (100%)	51 (46%)	61 (54%)
Blinded, n (%)	58 (51%)	17 (15%)	41 (36%)
Setting			
Firstline, n (%)	37 (33%)	22 (20%)	15 (13%)
Other, n (%)	75 (66%)	29 (25%)	46 (41%)
Drug class			
Chemotherapy, n (%)	4 (4%)	1 (1%)	3 (3%)
Endocrine Therapy, n (%)	9 (8%)	-	9 (8%)
Immunotherapy, n (%)	38 (34%)	26 (23%)	12 (11%)
Small Molecule, n (%)	50 (44%)	17 (15%)	33 (29%)
Antibody Therapy, n (%)	10 (9%)	7 (6%)	3 (3%)
Radiotherapeutics, n (%)	1 (1%)	-	1 (1%)
Indication Group			
Melanoma, n (%)	14 (12%)	8 (7%)	6 (5%)
Breast, n (%)	18 (16%)	6 (5%)	12 (11%)
GI (incl. HCC), n (%)	13 (12%)	7 (6%)	6 (6%)
Lung, n (%)	23 (20%)	15 (13%)	8 (7%)
Ovarian, n (%)	9 (8%)	-	9 (8%)
Prostate, n (%)	10 (9%)	-	10 (9%)
Renal, n (%)	8 (7%)	6 (5%)	2 (2%)
Urothelial, n (%)	3 (3%)	1 (1%)	2 (2%)
Sarcoma and GIST, n (%)	5 (4%)	4 (3%)	1 (1%)
Other, n (%)	9 (8%)	4 (4%)	5 (4%)

IQR: interquartile range, PFS: progression-free survival, OS: overall survival, GI: gastrointestinal, HCC: Hepatocellular Carcinoma, and GIST: Gastrointestinal Stroma Tumor.

**Table 2 cancers-17-01359-t002:** Subgroup analyses.

Subgroup	PFS HR (CI 95%) Restaging Interval < 8 Weeks	PFS HR (CI 95%) Restaging Interval ≥ 8 Weeks	*p* Value for the Subgroup Difference	Heterogeneity I^2^
All (n = 112)	0.58 (0.53, 0.63), (n = 51)	0.48 (0.44, 0.52), (n = 61)	0.004	90%
**Drug Class (n = 98)**				
Immunotherapy (n = 38)	0.70 (0.63, 0.77), (n = 26)	0.66 (0.57, 0.77), (n = 12)	0.62	83%
Small Molecules (n = 50)	0.43 (0.37, 0.50), (n = 17)	0.45 (0.41, 0.51), (n = 33)	0.63	81%
Antibodies (n = 10)	0.55 (0.43, 0.70), (n = 7)	0.55 (0.43, 0.70), (n = 3)	1.00	88%
**Indication Group (n = 76)**				
Breast (n = 18)	0.50 (0.39, 0.64), (n = 6)	0.58 (0.52, 0.65), (n = 12)	0.28	74%
Lung (n = 23)	0.60 (0.51, 0.69), (n = 15)	0.59 (0.46, 0.75), (n = 8)	0.93	87%
GI (incl. HCC) (n = 13)	0.57 (0.48, 0.68), (n = 7)	0.53 (0.46, 0.61), (n = 6)	0.52	79%
Melanoma (n = 14)	0.46 (0.36, 0.57), (n = 8)	0.63 (0.55, 0.73), (n = 6)	0.02	81%
Kidney (n = 8)	0.67 (0.55, 0.81), (n = 6)	0.44 (0.34, 0.58), (n = 2)	0.01	89%
Sarcoma and GIST (n = 5)	0.40 (0.1, 0.88), (n = 4)	0.31 (0.24, 0.40), (n = 1)	0.55	95%
**Trial Phase**				
Phase 3 (n = 105)	0.58 (0.52, 0.64), (n = 48)	0.49 (0.44, 0.53), (n = 57)	0.01	90%
Phase 2 (n = 7)	0.57 (0.49, 0.67), (n = 3)	0.40 (0.21, 0.74), (n = 4)	0.27	81%
**Primary Outcome**				
PFS (n = 55)	0.45 (0.39, 0.51), (n = 18)	0.45 (0.40, 0.51), (n = 37)	0.95	85%
OS (n = 25)	0.72 (0.63, 0.83), (n = 14)	0.58 (0.50, 0.67), (n = 11)	0.03	90%
PFS and OS (n = 25)	0.60 (0.52, 0.70), (n = 15)	0.52 (0.41, 0.65), (n = 10)	0.26	89%
Others (n = 7)	0.71 (0.50, 1.01), (n = 4)	0.35 (0.27, 0.44), (n = 3)	0.001	95%

GI: gastrointestinal, HCC: Hepatocellular Cancer, and GIST: Gastrointestinal Stroma Tumor.

**Table 3 cancers-17-01359-t003:** Sensitivity analyses of restaging intervals other than median.

Staging Interval (weeks)	HR for PFS 95% Cl	*p* for Subgroup Difference	Heterogeneity I^2^
<9 (n = 90) vs. ≥9 (n = 22)	0.54 (0.50, 0.58) vs. 0.45 (0.38, 0.54)	0.06	90%
<12 (n = 96) vs. ≥12 (n = 16)	0.55 (0.52, 0.59) vs. 0.38 (0.32, 0.44)	0.0001	90%
<8 (n = 48) vs. ≥8 (n = 57) without <6 and >12	0.60 (0.55, 0.66) vs. 0.49 (0.45, 0.54)	0.002	89%
≤6 (n = 51) vs. >6 to <12 (n = 45) vs. ≥12 (n = 16)	0.58 (0.53, 0.64) vs. 0.52 (0.49, 0.58) vs. 0.38 (0.32, 0.44)	<0.001	90%

## Data Availability

The original data presented in this study are openly available in FigShare at https://doi.org/10.6084/m9.figshare.28378277, accessed on 10 February 2025

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
