# Peer review of "The Effect of Staging Intervals on Progression-Free Survival in Registration Studies of Oncologic Drugs: A Meta-Analysis"

_cancers, 2025, doi:10.3390/cancers17081359_

Round 1
Reviewer 1 Report
Comments and Suggestions for Authors
The Author present a meta-analysis of the studies that supported the registration of cancer drugs in Switzerland from 2010 to 2022 in order to show wether the restaging interval during treatment might significantly influence treatment effect on PFS as evalutaetd by HR
The topic is of general interest in oncology, as no consensus exist on the optimal timing of patient re-evaluation during treatment.
The results study, showing higher HR linked to shorter intervals between scans (and thus an apparently lower effect of the compared treatments), reflect a low difference between treatment arms in terms of early relapse.
The paper is very well written ( please, check line 53. "...(PFS) has been adopted AS a primary outcome.." and results have been clearly and extensively described.
Author Response
Response to Reviewer 1 Comments
|
||
1. Summary |
|
|
Thank you very much for taking the time to review this manuscript. Please find the detailed responses below and the corresponding revisions/corrections highlighted/in track changes in the re-submitted files.
|
||
2. Questions for General Evaluation |
Reviewer’s Evaluation |
Response and Revisions |
Does the introduction provide sufficient background and include all relevant references? |
Yes |
We thank the reviewer for his/her assessment and favorable feedback. |
Are all the cited references relevant to the research? |
Yes |
|
Is the research design appropriate? |
Yes |
|
Are the methods adequately described? |
Yes |
|
Are the results clearly presented? |
Yes |
|
Are the conclusions supported by the results? |
Yes
|
|
3. Point-by-point response to Comments and Suggestions for Authors |
||
The Author present a meta-analysis of the studies that supported the registration of cancer drugs in Switzerland from 2010 to 2022 in order to show whether the restaging interval during treatment might significantly influence treatment effect on PFS as evaluated by HR. The topic is of general interest in oncology, as no consensus exist on the optimal timing of patient re-evaluation during treatment. The results study, showing higher HR linked to shorter intervals between scans (and thus an apparently lower effect of the compared treatments), reflect a low difference between treatment arms in terms of early relapse. The paper is very well written (please, check line 53. "...(PFS) has been adopted AS a primary outcome.." and results have been clearly and extensively described. Authors’ reply: We thank the reviewer for spotting the missing word which we have now added.
4. Response to Comments on the Quality of English Language
Reviewer’s Evaluation: The English is fine and does not require any improvement
|
Reviewer 2 Report
Comments and Suggestions for Authors
The meta-analysis addressed a prior reporting inverse relationship between tetsing interval and PFS HR of anticancer drugs. However the systematic review did not include any reference to methodology for conduct and reporting (e.g. PRISMA) and did not ascertain the study quality.
Moreover in the sbtract "oncologic drugs in palliative setting " is completely misleading
Thefore we suggest the authors to revise the manuscript according to PRISMA checklist and to asertain study quality by ROB2 or Newcastle tools
Author Response
Response to Reviewer 2 Comments
|
||
1. Summary |
|
|
Thank you very much for taking the time to review this manuscript. Please find the detailed responses below and the corresponding revisions/corrections highlighted/in track changes in the re-submitted files.
|
||
2. Questions for General Evaluation |
Reviewer’s Evaluation |
Response and Revisions |
Does the introduction provide sufficient background and include all relevant references? |
Can be improved |
Please see replies below where we address the point brought up. |
Are all the cited references relevant to the research? |
Can be improved |
|
Is the research design appropriate? |
Can be improved |
|
Are the methods adequately described? |
Can be improved |
|
Are the results clearly presented? |
Can be improved |
|
Are the conclusions supported by the results? |
Can be improved |
|
3. Point-by-point response to Comments and Suggestions for Authors |
||
The meta-analysis addressed a prior reporting inverse relationship between testing interval and PFS HR of anticancer drugs. However, the systematic review did not include any reference to methodology for conduct and reporting (e.g. PRISMA) and did not ascertain the study quality. Thefore we suggest the authors to revise the manuscript according to PRISMA checklist and to asertain study quality by ROB2 or Newcastle tools.
Authors’ reply: The PRISMA reporting guidelines were developed to allow a structured conduct and reporting of systematic reviews and meta-analyses and have been used be authors in many other publications. However, the manuscript under review is not a systematic review (which would also require a detailed reporting of the search strategy including MeSH terms and searched databases) while we had not referred to the PRISMA reporting guideline. However, given that we indeed did a meta-analysis, we agree with the reviewer that the PRISMA guideline applies. We have modified the manuscript accordingly and submit the complete PRISMA guideline as Supplement.
Given the nature of the included studies (i.e. registration studies) we argue that formal assessment of study quality using the Newcastle-Ottawa Scale and Rob2 tool for risk of bias assessment does not add relevant information since the quality of the studies has been rated by the national authorization and supervisory authority for drugs and medical products as high enough to allow drug approval based on the respective studies. Whether the quality of the studies would also fulfill quality criteria according to above mentioned tools is beyond the scope of our research question. Therefore, we opt not to apply the ratings in the specific context of the work under review and have now addressed this question in the revised version of the manuscript.
Moreover, in the abstract "oncologic drugs in palliative setting " is completely misleading.
Authors’ reply: We have now modified the wording to “drugs to treat incurable solid tumors” |
||
|
||
|
||
|
||
4. Response to Comments on the Quality of English Language
Reviewer’s Evaluation: The English is fine and does not require any improvement.
|
||
|
Reviewer 3 Report
Comments and Suggestions for Authors
The inclusion criteria excluded pediatric studies, and hematologic cancers, potentially leading to selection bias that affects the generalizability of the results. The study reports significant heterogeneity, yet it does not clearly justify how this was handled. It assumes that staging intervals directly impact PFS but does not explore whether factors like disease biology, prior treatments, or imaging protocols could be influencing these results.
Comments on the Quality of English Languageminor revision
Author Response
Response to Reviewer 3 Comments
|
||
1. Summary |
|
|
Thank you very much for taking the time to review this manuscript. Please find the detailed responses below and the corresponding revisions/corrections highlighted/in track changes in the re-submitted files.
|
||
2. Questions for General Evaluation |
Reviewer’s Evaluation |
Response and Revisions |
Does the introduction provide sufficient background and include all relevant references? |
Must be improved |
Please see replies below where we address the point brought up. |
Are all the cited references relevant to the research? |
Must be improved |
|
Is the research design appropriate? |
Must be improved |
|
Are the methods adequately described? |
Must be improved |
|
Are the results clearly presented? |
Must be improved |
|
Are the conclusions supported by the results? |
Must be improved
|
|
3. Point-by-point response to Comments and Suggestions for Authors |
||
The inclusion criteria excluded pediatric studies, and hematologic cancers, potentially leading to selection bias that affects the generalizability of the results. The study reports significant heterogeneity, yet it does not clearly justify how this was handled. It assumes that staging intervals directly impact PFS but does not explore whether factors like disease biology, prior treatments, or imaging protocols could be influencing these results. Authors’ reply: We thank the reviewer for the critical comments. As pointed out studies supporting approvals of drug used in pediatric oncology and hematologic malignancies were excluded. The reason for this was that in the pediatric setting only a very small number of drugs were newly approved and that in hematologic malignancies staging intervals in the non-curative setting (e.g. low grade lymphoma) are typically longer than in solid tumors treated with palliative intent or are not staged using imaging (e.g. myeloma, myeloproliferative diseases). However, we have now added the exclusion of studies supporting drug approvals of drug used in pediatric oncology and hematologic malignancies as further potential limitation. For good reasons the reviewer highlights the presence of significant heterogeneity. As stated in the manuscript this could not be explained by outlier studies and might be among the reasons why no clear impact of staging intervals was observed. Indeed, we hypothesized that shorter staging intervals impact PFS but did not find that this was the case. As such, the hypothesis could be rejected. We agree that the exploration of other factors potentially impacting PFS results would be of interest but argue that such analyses should be driven be a-priori formulated hypotheses and are beyond the scope of our work.
4. Response to Comments on the Quality of English Language
Reviewer’s Evaluation: minor revision
Authors’ reply: We have once more carefully reviewed the language which was assessed as fine by two other reviewers without need for improvement.
|
||
|
Round 2
Reviewer 3 Report
Comments and Suggestions for Authors
The meta-analysis reports extreme heterogeneity. Grouping studies into <8 vs. ≥8 weeks oversimplifies the relationship. While sensitivity analyses use other cut-offs, the primary analysis relies on a median split, potentially misrepresenting continuous effects. Contradictory findings like melanoma vs. kidney cancer lack biological or methodological explanations. The authors do not explore whether tumor biology or assessment protocols drive these discrepancies. The discussion fails to contextualize the observed HR differences. Are these statistically significant differences clinically meaningful? Including phase 2 trials with phase 3 trials introduces bias. No sensitivity analysis excludes phase 2 studies to test robustness. Conduct subgroup analyses stratified by trial phase, region, and assessment protocols to address heterogeneity. Perform meta-regression to adjust for covariates
Author Response
Response to Reviewer, Round 2 Comments
|
||
1. Summary |
|
|
We want to express our gratitude for taking the time to review the revised manuscript for the 2nd round. Please find the detailed responses below and the corresponding revisions/corrections highlighted/in track changes in the re-submitted files.
|
||
2. Questions for General Evaluation |
Reviewer’s Evaluation |
Response and Revisions |
Does the introduction provide sufficient background and include all relevant references? |
Must be improved |
Please see replies below where we address the point brought up. |
Are all the cited references relevant to the research? |
Must be improved |
|
Is the research design appropriate? |
Must be improved |
|
Are the methods adequately described? |
Must be improved |
|
Are the results clearly presented? |
Must be improved |
|
Are the conclusions supported by the results? |
Must be improved
|
|
3. Point-by-point response to Comments and Suggestions for Authors |
||
The meta-analysis reports extreme heterogeneity. Grouping studies into <8 vs. ≥8 weeks oversimplifies the relationship. While sensitivity analyses use other cut-offs, the primary analysis relies on a median split, potentially misrepresenting continuous effects. Contradictory findings like melanoma vs. kidney cancer lack biological or methodological explanations. The authors do not explore whether tumor biology or assessment protocols drive these discrepancies. The discussion fails to contextualize the observed HR differences. Are these statistically significant differences clinically meaningful? Including phase 2 trials with phase 3 trials introduces bias. No sensitivity analysis excludes phase 2 studies to test robustness. Conduct subgroup analyses stratified by trial phase, region, and assessment protocols to address heterogeneity. Perform meta-regression to adjust for covariates.
Authors’ reply: We thank the reviewer for the his/her feedback, comments and inputs. We do agree that heterogeneity warrants further exploration and also the contradictory findings for melanoma vs. kidney cancer. Of note, high heterogeneity was anticipated given that the studies pooled in our analysis had considerable clinical heterogeneity. This was considered in our predefined statistical plan which specified exploratory subgroup and sensitivity analyses and which also specified to use the median of staging intervals as primary cut-off (as done in the cited work by Dabush and colleagues we aimed to corroborate).
With regards to heterogeneity we have now run further subgroups analyses according to the phase of the studies and also according to the primary outcome of the respective studies as suggested (highlighted in Table 2). We fully agree that the suggested subgroup analyses according to region would be of great interest (indeed our group published a meta-analysis on this topic, https://pubmed.ncbi.nlm.nih.gov/26542275/ ) but this was not feasible here since many of the registration studies included in our analysis enrolled patients from various regions around the world and did not report sufficient outcomes, if at all, according to regions.
We also did further sensitivity analyses as recommended, specifically repeating the analyses after exclusion of phase 2 studies. The results are presented in a supplement (Table S1) and are in line with the main analysis. Also, we have further explored whether removal of outlier studies might explain heterogeneity which was not the case.
We have now repeated a meta-regression analyses with staging intervals as continuous variable to explore a potential impact on HRs for PFS. Here, a moderate correlation was observed, in line with the finding when using a binary cut-off. A sensitivity analyses after exclusion of phase 2 studies was also conducted and is described in the manuscript. However, we opted not to do a formal multivariable regression analysis since we lack adequate power to yield meaningful results (not suffering from overfitting). Yet, this may certainly be considered a shortcoming and we have now also mentioned this as a limitation.
Last but not least the question of question of contradictory findings for melanoma vs. kidney cancer remains to be answered (likewise why the observed effect appears to be driven by studies with OS as primary outcome, a new finding arising from the additional subgroup analyses described above). We have tried to further address this question in the discussion but these thoughts are hypothesis-generating, at best. We hypothesis that tumours with more aggressive biology may exhibit shorter time to progression, leading to more frequent assessment. Since changes in PFS over brief intervals could have the tendency to be modest, this might influence an apparent effect on hazard ratio observed in such studies. However, given the small numbers of in the subgroup of studies in melanoma, it may well be that this finding is just to due chance, i.e. not real. Indeed, the practical relevance of staging interval on HRs for PFS remains open – we started our work with the idea to corroborate an earlier observation that shorter staging intervals are associated with lower HRs. This hypothesis could be rejected which seems reassuring. We remain puzzled by the finding that the overall studies with OS as primary outcome appear to drive the overall effect. However, even if this finding was real, it would not be of great relevance since OS remains are more important outcome for patients than any PFS measure. We have also added this to the discussion. |
||
|
Round 3
Reviewer 3 Report
Comments and Suggestions for Authors
significant improved after revision
Author Response
We thank the reviewer for his/her time and valuable feedback.